

# Using algorithmic trading to analyze short term profitability of Bitcoin

Iftikhar Ahmad[1], Muhammad Ovais Ahmad[2,3], Mohammed A. Alqarni[5], Abdulwahab Ali Almazroi[4] and Muhammad Imran Khan Khalil[1]

[1] Department of Computer Science and Information Technology, University of Engineering & Technology Peshawar, Peshawar, Pakistan
[2] Department of Mathematics and Computer Science, Karlstad University, Karlstad, Sweden
[3] M3S Research Unit, University of Oulu, Oulu, Finland
[4] University of Jeddah, College of Computing and Information Technology at Khulais, Department of Information Technology, Jeddah, Saudi Arabia
[5] University of Jeddah, College of Computer Science and Engineering, Department of Software Engineering, Jeddah, Saudi Arabia

## ABSTRACT

Cryptocurrencies such as Bitcoin (BTC) have seen a surge in value in the recent past and appeared as a useful investment opportunity for traders. However, their short term profitability using algorithmic trading strategies remains unanswered. In this work, we focus on the short term profitability of BTC against the euro and the yen for an eight-year period using seven trading algorithms over trading periods of length 15 and 30 days. We use the classical buy and hold (BH) as a benchmark strategy. Rather surprisingly, we found that on average, the yen is more profitable than BTC and the euro; however the answer also depends on the choice of algorithm. Reservation price algorithms result in 7.5% and 10% of average returns over 15 and 30 days respectively which is the highest for all the algorithms for the three assets. For BTC, all algorithms outperform the BH strategy. We also analyze the effect of transaction fee on the profitability of algorithms for BTC and observe that for trading period of length 15 no trading strategy is profitable for BTC. For trading period of length 30, only two strategies are profitable.

## INTRODUCTION

Cryptocurrencies have seen a surge in the recent past. Researchers and investors alike have focused on the growth and evolution of cryptocurrencies like Bitcoin (*BTC*), Etherum, and Litecoin etc. *Moore (2013)* attributed three main factors that contributed towards the rise and adaptation of bitcoins. First, higher profit margins, maintained by credit card agencies for using their platforms has resulted in dis-satisfied customers. The customers are thus lured to use *BTC*, which promises extremely low transaction fee. Second, the anonymity that is offered by the bitcoins. Bitcoins offer the possibility of conducting transactions using pseudonyms and thus omitting the need of using real names. Third is the decentralization of the bitcoin that protects against inflation. Over time, *BTC* has become one of the choice currencies for online payment and beside others is accepted by tech-giants like Amazon, Apple, Microsoft, and Paypal etc. The introduction of cryptocurrencies provided a new

Corresponding author
Iftikhar Ahmad,
ia@uetpeshawar.edu.pk

investment domain for the investors, and became a credible investment vehicle (*Brière, Oosterlinck & Szafarz, 2015*).

Bitcoin was introduced by Satoshi Nakamoto in 2008 (*Nakamoto, 2008*). The inventor Satoshi Nakamoto is a pseudonym and the real identity of the person is not known to the world. Bitcoin is a digital currency, i.e., unlike fiat currencies such as dollar, and Euro etc., it does not have any physical denomination, and is present only in digital form. Beside similarities, such as the price regulation based on demand and supply, there are some key differences between fiat and crypto currencies like $BTC$. For instance, $BTC$ has no centralized authority (like the Federal Reserve) that controls the supply, i.e., $BTC$ and by extension all cryptocurrencies are decentralized by nature. The value of a fiat currency is generally dependent on factors such as inflation rate in a country, the interest rates, balance between import and export and monetary policy. In contrast, the value of $BTC$ can be determined by several factors such as transactional demand, media speculation, buzz around the technology, and acceptability etc (*Nguyen, de Bodisco & Thaver, 2018*; *Wang & Vergne, 2017*). Other differentiating aspects include legality, tangibility, and storage.

The underlying technology of $BTC$ is blockchain. In its simplest form, a blockchain is a distributed append-only ledger formed by the collection of blocks. The append-only nature of the ledger means that transactions once recorded are tempered-proof and cannot be changed/modified in any form. This property is achieved with the help of cryptographic hash functions (*Narayanan et al., 2016*). The Bitcoin eco-system is based on peer-to-peer network where a large number of computational nodes are connected (not necessarily directly). The peer-to-peer network omits the need of centralized system, instead it uses the concept of "proof-of-work" to validate transactions. For a detailed description of BTC, its underlying technology and applications, the reader is referred to *Narayanan et al. (2016)*.

Algorithmic trading is an important tool used by investors in financial trading markets (*Ahmad & Schmidt, 2012*). It facilitates investors in investing their wealth in various assets (currencies, bonds, stock shares etc.) by automating the decision making process. A number of algorithms are proposed in the literature for algorithmic trading (*Iqbal & Ahmad, 2015*; *Mohr, Ahmad & Schmidt, 2014*; *Ahmad & Schmidt, 2012*; *El-Yaniv et al., 2001*). The problem is addressed in a wide variety of domains including computer science (*Kao & Tate, 1999*; *El-Yaniv et al., 2001*; *Mohr, Ahmad & Schmidt, 2014*), operations research (*Schroeder, Dochow & Schmidt, 2018*), economics and finance (*Coakley, Marzano & Nankervis, 2016*; *Hsu, Hsu & Kuan, 2010*). These algorithms are based on various assumptions and are designed to optimize a variety of objective functions such as minimizing competitive ratio (*Mohr, Ahmad & Schmidt, 2014*).

Algorithmic trading and technical analysis are also important tools to investigate the market behavior and assess its profitability in the short and long term scenarios (*Ahmad & Schmidt, 2012*; *Coakley, Marzano & Nankervis, 2016*). Despite the debate in the literature questioning the effectiveness of technical analysis, there is a plethora of research work based on technical analysis (*Coakley, Marzano & Nankervis, 2016*; *Hsu, Hsu & Kuan, 2010*; *Menkhoff & Taylor, 2007*). The variety of studies validated the usefulness of technical analysis and its wide spread applicability. However, to the best of our knowledge, there is no work to evaluate the short term profitability of $BTC$ using algorithmic trading

and technical analysis. We investigate the short term profitability of *BTC* against two other major currencies euro and yen. More specifically, we consider daily exchange rates of dollar–BTC, dollar–euro, and dollar-yen from 1st Jan 2011 to 31st Dec 2018. We investigate the short term profitability (15 and 30 days) as it is a common observation that in the long term *BTC* has observed significant price movement and is highly profitable.

We consider two categories of algorithms namely reservation price algorithms and moving average based algorithms and consider buy and hold as a benchmark strategy. Our findings are based on the geometric average period return, the effect of transaction fee, the number of buy and sell transactions, the number of completed transactions, and the number of profitable vs. non-profitable transactions. Using buy and hold strategy as our benchmark, we compare the geometric average period returns of the seven strategies with buy and hold. Rather surprisingly, we found that in short term the yen is more profitable than *BTC* and euro; however the answer also depends on the choice of algorithm. Reservation price algorithms result in 7.5% and 10% of average returns over 15 and 30 days respectively, which is the highest for all algorithms for the three assets. For *BTC*, all algorithms outperform the *BH* strategy. After introducing a transaction fee of 4%, we observe that for the trading period of length 15 no trading strategy is profitable for *BTC*, whereas for trading period of length 30, only two strategies are profitable.

It is important to mention that we do not consider machine learning techniques but instead focus on algorithmic trading strategies which do not rely on past trends and patterns, thus do not need future to follow the patterns of the past. Machine learning based algorithms are presented in the literature, the reader is referred to *Uras et al. (2020)*, *Alessandretti et al. (2018)* and *Zbikowski (2016)*

Rest of the paper is organized as follows; In 'Literature Review', we briefly present literature review on the use of experimental evaluation of trading algorithms. In 'Research Questions and Data Set', we present a set of research questions and the methodology for the extraction of data set. In 'Experimental Setup and Methodology', we describe the set of algorithms, followed by the description of the evaluation criterion. Results are presented in 'Results and Discussions', whereas 'Conclusion' presents conclusion, and directions for future work.

## LITERATURE REVIEW

Experimental evaluation of trading strategies is an established area of research in Computer Science (*Iqbal, Ahmad & Schmidt, 2012*; *Ahmad & Schmidt, 2012*; *Mohr, Ahmad & Schmidt, 2014*), and Computational Finance (*Brock, Lakonishok & LeBaron, 1992*; *Coakley, Marzano & Nankervis, 2016*). Ever since the seminal work of *Brock, Lakonishok & LeBaron (1992)* there is a considerable literature devoted to the study of algorithmic trading strategies. The strategies are investigated from different perspectives and for various markets around the world. In the following, we present a brief literature review of the work based on experimental analysis of trading algorithms.

*Iqbal, Ahmad & Schmidt (2012)* performed an experimental evaluation of *DAX* 30 to answer the question "Can online trading algorithms beat the market?". The authors

considered a number of trading algorithms, and compared their performance with classical buy and hold (*BH*) algorithm over trading periods of various length. They concluded that trading algorithms can beat the market, i.e., a trading algorithm can achieve a better return (profit) than *BH* algorithm. *Ahmad & Schmidt (2012)* presented an extensive experimental evaluation of trading algorithms for uni-directional conversion problem (see *Mohr, Ahmad & Schmidt (2014)* for a definition of uni-directional conversion problem). The authors considered two data sets *DAX*30 and *S&P*500 over a period of 10 years (2001–2010, and compared the performance of various algorithms using average competitive ratio. Unlike, *Iqbal, Ahmad & Schmidt (2012)*, *Ahmad & Schmidt (2012)* used bootstrapping to avoid data snooping bias.

*Coakley, Marzano & Nankervis (2016)* performed a comprehensive analysis of various trading rules for 22 currencies over a period of 19 years. Authors reported evidence of profitability for rules based on classical moving average as well as rules based on Bollinger bands and relative strength index. *Jiang, Tong & Song (2019)* investigated the profitability of trading rules in Chinese stock market. The authors used 19 years daily data from Chinese aggregate market return and confirmed the profitability of trading rules even in the presence of transaction costs. *Strobel & Auer (2018)* analyzed the diminishing predictive power of fundamental variables and seasonal effects over time. They considered Variable Length Moving Average (*VLMA*) rules introduced by *Brock, Lakonishok & LeBaron (1992)*, and using data set covering 1972 to 2015 concluded that *VLMA* rules have lost the predictive ability. *Chang, Jong & Wang (2017)* used *VLMA* rules to Taiwanese Stock Exchange (*TWSE*) and computed excess returns to buy and hold (*BH*) strategy. The objective of the work was to evaluate the effectiveness of *VLMA* rules against *BH*. The results confirmed the superiority of *VLMA* rules against *BH*. The novelty of the work lies in the application of *VLMA* rules to all individual stock listed on *TWSE*. *Hsu, Taylor & Wang (2016)* investigated the profitability of technical trading rules in the forex market by analyzing 30 currencies over a period of 45 years. It is argued that there is a significant evidence of the profitability of technical trading rules for some periods. Likewise, the profitability variations are consistent with the adaptive market hypothesis. *Fang, Jacobsen & Qin (2014)* used the technical trading rules of *Brock, Lakonishok & LeBaron (1992)* and out-of-sample tests based on fresh data. They inferred that there is no conclusive evidence to support the predictive ability of these strategies. However, they attributed the lack of predictive ability to potential bias rather than efficient market hypothesis.

Despite the plethora of work dedicated to analyze the profitability of various trading strategies and markets, to the best of our knowledge there is no work that compares the profitability of Bitcoin with various other currencies, and to evaluate the performance of various algorithms on Bitcoin.

## RESEARCH QUESTIONS AND DATA SET

### Research questions

We formulate a set of research questions (RQ), which essentially provide a base for the data analysis. The main objectives of the research questions are to identify the most profitable

asset, the most appropriate (profitable) algorithm for various assets, and to analyze the effect of transaction cost on the profitability of various algorithms.

RQ 1. Which asset is the most profitable in terms of geometric average period returns?

RQ 2. Which strategy is the most profitable for each of the assets?

RQ 3. How the number of buy and sell signals vary for *BTC*?

RQ 4. What are the number of positive and negative returned transactions for *BTC*?

RQ 5. How the transaction fee effects the profitability of algorithms?

Note that the research questions are not arbitrarily but are instead rooted in the literature. For instance, *RQ1* is based on *Hsu, Taylor & Wang (2016)* who used Japanese yen, German mark/euro, U.K. pound, and Swiss franc as base currency in their study and evaluated the profitability of technical trading rules. Likewise, *RQ2* is variant of research question posed in *Abbey & Doukas (2012)*. In *Abbey & Doukas (2012)* the authors examined if technical trading rules can be profitable for individual traders. In the similar manner, *RQ5* is studied by a number of researchers including *Hsu, Taylor & Wang (2016)* and *Ahmad & Schmidt (2012)*.

## Data

We consider the daily closing prices of the following currencies against dollar;

  i  Bitcoin (BTC)

 ii  Euro

iii  Yen

A single data point represents the amount of currency that can be purchased by spending 1 *US*. The BTC data is obtained from coindesk website (http://www.coindesk.com) for a period of 8 years starting from 1 Jan 2011 to 31 Dec 2018. The main reason for the selection of the data set is based on the availability of the data. On many websites such as coindesk, USD-BTC data is only available from 18 July 2010, therefore, we select the starting date to be 1 Jan 2011, and in the process data set consists of complete 8 years. Euro and yen data is obtained from Yahoo! Finance (http://finance.yahoo.com).

Table 1 reports various statistics for the data. For the sake of comparison, we take a holistic view of the whole data set by reporting the statistics for the 8 years (1 Jan 2011–31 Dec 2018).

## EXPERIMENTAL SETUP AND METHODOLOGY

### Trading algorithms

A variety of trading algorithms are proposed in the literature (*Mohr, Ahmad & Schmidt, 2014*; *Coakley, Marzano & Nankervis, 2016*). In the following we describe a selected set of algorithms that are used in our study. The motivation behind the selection of the algorithms from the literature is rooted in the performance of the algorithms. Studies (*Ahmad & Schmidt, 2012*; *Iqbal, Ahmad & Schmidt, 2012*; *Iqbal, Ahmad & Shah, 2019*) have shown that reservation algorithm of *El-Yaniv et al. (2001)* and *Iqbal, Ahmad & Shah (2019)* are the best performing algorithms. Further, in order to make the comparison meaningful, two widely used techniques from finance namely *variable length moving average*, and *fixed length moving average* are also considered.

**Table 1  Summary Statistics of the Dataset, $\sigma$ = Standard Deviation, $\gamma$ = Skewness, $\mathcal{K}$ = Kurtosis, $\rho(k) = k$th order correlation.**

| Asset | Bitcoin | Euro | Yen |
|---|---|---|---|
| Observations | 2921 | 2086 | 2086 |
| Minimum | 0.0000517 | 0.674 | 75.82 |
| Maximum | 3.448 | 0.817 | 125.62 |
| Mean | 0.099 | 0.818 | 101.94 |
| $\sigma$ | 0.345 | 0.075 | 14.62 |
| $\gamma$ | 6.234 | 0.0703 | −0.475 |
| Std Error of $\gamma$ | 0.0452 | 0.0535 | 0.0535 |
| $\mathcal{K}$ | 45.178 | −1.327 | −1.069 |
| Std Error of $\mathcal{K}$ | 0.0905 | 0.1071 | 0.1071 |
| $\rho(1)$ | 0.9984 | 2.9856 | 375.7 |
| $\rho(2)$ | 0.9969 | 2.9820 | 375.2 |
| $\rho(3)$ | 0.9953 | 2.9785 | 374.7 |
| $\rho(4)$ | 0.9937 | 2.9752 | 374.3 |
| $\rho(5)$ | 0.9922 | 2.9790 | 373.8 |
| $\rho(6)$ | 0.9906 | 2.9684 | 373.4 |
| $\rho(7)$ | 0.9890 | 2.9651 | 372.9 |

### Reservation price algorithms

Reservation price algorithm calculates a threshold price and generates a buy signal when the offered exchange rate is less than or equal to the threshold. A sell signal is generated when the price is at least threshold (*Iqbal, Ahmad & Schmidt, 2012*; *Iqbal, Ahmad & Shah, 2019*; *Kao & Tate, 1999*). A number of reservation price algorithms are presented in the literature (*Mohr, Ahmad & Schmidt, 2014*; *Kao & Tate, 1999*; *Iqbal, Ahmad & Shah, 2019*; *El-Yaniv et al., 2001*). In the following we present the selected set of reservation price algorithms considered for our study.

*El-Yaniv et al. (2001)* assumed a priori information about the lower (minimum possible price *m*) and upper (maximum possible price *M*) bound of prices, and presented a reservation price algorithm. Let $e_t$ be the current exchange price. Algorithm 1 provides formal description for El-Yaniv reservation price algorithm for generating buy and sell signals respectively.

*Iqbal, Ahmad & Shah (2019)* presented a modified version of the reservation price policy of *El-Yaniv et al. (2001)*. The authors critiqued the assumption of fixed values of *m* and *M* and argued that inter-day price fluctuation is not arbitrary but is instead governed by inter-day price fluctuation function as shown in Eq (1).

$$(1 - \gamma)e_{t-1} \leq e_t \leq (1 + \gamma)e_{t-1} \tag{1}$$

Note that $e_t$ is the exchange rate offered on day $t$, and $\gamma \in 0, 1$. The formal description of algorithm is given in Algorithm 2. For detailed working of the algorithm, the reader is referred to *Iqbal, Ahmad & Shah (2019)*.

*Kao & Tate (1999)* presented a reservation price algorithm based on the perceived rank of the offered exchange rate. The perceived exchange rank is calculated based on the current

---

**Algorithm 1** Reservation Price Algorithm ($RP$)

---

**Require:** $e_t, m, M$

  1: Calculate reservation price $e^* = \sqrt{Mm}$

  2: **if** $e_t \leq e^*$ **then**

  3:     Generate a buy signal

  4: **end if**

  5: **if** $e_t > e^*$ **then**

  6:     Generate a sell signal

  7: **end if**

  8: Generate a sell signal on the last trading day if there is an open buy signal even if the criterion for sell signal is not met.

---

**Algorithm 2** Reservation Price Algorithm $RP^*$

---

**Require:** $m, M, \gamma, T$

  1: Set $m_0 = m, M_0 = M$

  2: **for** t=1 to T **do**

  3:     A new exchange rate $e_t$ is observed.

  4:     Compute achievable lower bound $m_t$:
$$m_t = \max\{m_{t-1}, e_t(1-\gamma)^{T-t}\}$$

  5:     Compute achievable upper bound $M_t$:
$$M_t = \min\{M_{t-1}, e_t(1+\gamma)^{T-t}\}$$

  6:     Calculate new reservation price $e_t^* = \sqrt{m_t M_t}$

  7:     **if** $e_t \leq e_t^*$ **then**

  8:         Generate a buy signal

  9:     **end if**

10:     **if** $e_t > e_t^*$ **then**

11:         Generate a sell signal

12:     **end if**

13: **end for**

14: Generate a sell signal on the last trading day if there is an open buy signal even if the criterion for sell signal is not met.

---

rank $x_t$ of the offered exchange rate $e_t$ in all the exchange prices observed so far. The formal algorithm is presented in Algorithm 3.

$T$ represents the number of days in a trading period, $\mathcal{L}_T(t)$ and $\mathcal{H}_T(t)$ are the thresholds for buy and sell signals respectively and are computed as shown in Eqs. (2) and (4) respectively;

$$\mathcal{L}_T(t) = \begin{cases} 0 & : t = T \\ \left\lfloor \frac{t+1}{T+1}(R_T(t+1) - P_T(T+1)) \right\rfloor & : t < T \end{cases} \tag{2}$$

---

**Algorithm 3** Reservation Price Algorithm $(KT)$

---

**Require:** $e_t, \mathcal{L}_T(t), \mathcal{H}_T(t)$

Calculate $\mathcal{L}_T(t)$

Calculate $\mathcal{H}_T(t)$

Generate a buy signal at exchange rate $e_t$ if $x_t \leq \mathcal{L}_T(t)$

Generate a sell signal at exchange rate $e_t$ if $x_t \geq \mathcal{H}_T(t)$

Generate a sell signal on the last trading day if there is an open buy signal even if the criterion for sell signal is not met.

---

Note that $P_T(t)$ is the expected difference between the buy and sell prices if the optimal strategy is followed at $t$, and is calculated as shown in Eq. (3).

$$P_T(t) = \begin{cases} 0 & : t = T \\ P_T(t+1) + \dfrac{\mathcal{L}_T(t)}{t}\left(R_T(t+1) - P_T(t+1) - \dfrac{T+1}{t+1}\dfrac{\mathcal{L}_T(t)+1}{2}\right) & : t < T \end{cases} \tag{3}$$

$$\mathcal{H}_T(t) = \left\lceil \frac{t+1}{T+1} R_T(t+1) \right\rceil \tag{4}$$

Note that $R_T$ is the expected final rank of $e_t$ for selling, if an optimal strategy is followed starting from time $t$, and is calculated as given in Eq. (5).

$$R_T(t) = \frac{\mathcal{H}_T(t)-1}{t}\left(R_t(t+1) - \frac{T+1}{2(t+1)}\mathcal{H}_T(t)\right) + \frac{T+1}{2}. \tag{5}$$

### Moving average based rules

Moving average $(MA)$ rule is the simplest and popular technical analysis trading rule. The basic idea of $MA$ based rules is to generate buy and sell signals based on the short vs long-term moving averages. More specifically, a buy signal is generated when the short term moving average cuts the long term moving average from below. On the contrary, a sell signal is generated when the short-term moving average cuts the long-term moving average from above. However, in a market the crossing between short-term and long-term moving averages can occur on multiple instances in a short period, resulting in a large number of buy and sell signals (*Zhu et al., 2015*). The resulting large number of signals are hardly profitable and can force a large transaction fee as well. To avoid this, a minimum threshold called band is introduced. The band introduces a specific percentage difference between the short and long term moving averages in order to generate buy and sell signals.

In the literature two variants of the moving averages, called Variable Length Moving Average (*VLMA*) and Fixed Length Moving Average (*FLMA*) are used (*Brock, Lakonishok & LeBaron, 1992*; *Gunasekarage & Power, 2001*; *Zhu et al., 2015*). Let $\mathcal{A}_S$ be the short term moving average, $\mathcal{A}_L$ be the long term moving average, and $\mathcal{B}$ the band value. Algorithm 4 describes variable length moving average strategy for buy and sell signals.

In *VLMA* a buy signal is generated when the short term moving average cuts the long term moving average (taking into account the band value) from below, i.e., $\mathcal{A}_S > (1 + \mathcal{B})\mathcal{A}_L$.

---

**Algorithm 4** Variable Length Moving Average Algorithm

---

**Require:** $\mathcal{A_S}, \mathcal{A_L}, \mathcal{B}$

1: **if** $\mathcal{A_S} > (1+\mathcal{B})\mathcal{A_L}$ **then**

2:  Generate a buy Signal

3: **end if**

4: **if** $\mathcal{A_S} < (1-\mathcal{B})\mathcal{A_L}$ **then**

5:  Generate a sell signal

6: **end if**

7: Generate a sell signal on the last trading day if there is an open buy signal even if the criterion for sell signal is not met.

---

Likewise, *VLMA* generates a sell signal when $\mathcal{A_S} < (1-\mathcal{B})\mathcal{A_L}$. A rule is represented by the combination of three values $\mathcal{S}$ (length of short-term moving average), $\mathcal{L}$ (length of long-term moving average), $\mathcal{B}$ (band). For instance, $VLMA(5, 30, 0.02)$ represents a rule where short term average is taken over a period of 5 days, long term over a period of 30 days, and the band value is 2%. *FLMA* works on the same principle as stated in Algorithm 4. However, *FLMA* differs from the *VLMA* by introducing a holding period, i.e., once a signal is generated then the position must be held for a fixed number of days. Any signal generated during the holding period is ignored.

### Buy and hold strategy

Buy and hold (*BH*) is widely used in the literature as a benchmark strategy (*Mohr, Ahmad & Schmidt, 2014*; *Chang, Jong & Wang, 2017*; *Baur et al., 2018*), and is therefore used in our study as well. In *BH* an investor executes the buy transaction on the first day of the investment period and holds the position until the last day $T$. On the last day, a sell transaction is executed to complete the trading. The formal description of buy and sell signals of *BH* algorithm is given in Algorithm 5.

---

**Algorithm 5** Buy and Hold (*BH*)

---

**Require:** $e_1, e_T$

1: Buy on the first offered exchange rate $e_1$

2: Sell on the last offered exchange rate $e_T$

---

We test the profitability of the algorithms for various parameters and for various durations. We consider short term moving averages over 5 and 10 days, long term moving averages over 15, and 30 days, and band values 0.01 and 0.02 (*Brock, Lakonishok & LeBaron, 1992*; *Fang, Jacobsen & Qin, 2014*). Thus we produce a total of 8 trading rules, 4 each for *VLMA* and *FLMA*. Likewise, we consider 15 and 30 days durations for Algorithms 1, 2, 3 and 5. Thus we have a total of 16 variants of algorithms to evaluate. Table 2 presents a summary of the selected algorithms and their variants.

### Evaluation criterion

We use *Geometric Average Trading Period Return* (*GPR*) as our evaluation criterion. *GPR* is used as an evaluation criterion in a number of works such as *Schmidt, Mohr & Kersch*

**Table 2  Summary of the selected algorithms and their variants.**

| S.No | Algorithm | Description |
|------|-----------|-------------|
| 1 | $VLMA(5, 15, 0.01)$ | $VLMA$ algorithm with $\mathcal{S}, \mathcal{L}, \mathcal{B}$ values as $5, 15, 0.01$ |
| 2 | $VLMA(5, 15, 0.02)$ | $VLMA$ algorithm with $\mathcal{S}, \mathcal{L}, \mathcal{B}$ values as $5, 15, 0.02$ |
| 3 | $FLMA(5, 15, 0.01)$ | $FLMA$ algorithm with $\mathcal{S}, \mathcal{L}, \mathcal{B}$ values as $5, 15, 0.01$ |
| 4 | $FLMA(5, 15, 0.02)$ | $FLMA$ algorithm with $\mathcal{S}, \mathcal{L}, \mathcal{B}$ values as $5, 15, 0.02$ |
| 5 | $RP(15)$ | Reservation price algorithm ($RP$) (*El-Yaniv et al., 2001*) applied over 15 days |
| 6 | $RP^*(15)$ | Update reservation price algorithm ($RP^*$) (*Iqbal, Ahmad & Shah, 2019*) applied over 15 days |
| 7 | $KT(15)$ | Reservation price algorithm ($KT$) (*Kao & Tate, 1999*) applied over 15 days |
| 8 | $BH(15)$ | Buy and Hold algorithm applied over 15 days |
| 9 | $VLMA(10, 30, 0.01)$ | $VLMA$ algorithm with $\mathcal{S}, \mathcal{L}, \mathcal{B}$ values as $10, 30, 0.01$ |
| 10 | $VLMA(10, 30, 0.02)$ | $VLMA$ algorithm with $\mathcal{S}, \mathcal{L}, \mathcal{B}$ values as $10, 30, 0.02$ |
| 11 | $FLMA(10, 30, 0.01)$ | $FLMA$ algorithm with $\mathcal{S}, \mathcal{L}, \mathcal{B}$ values as $10, 30, 0.01$ |
| 12 | $FLMA(10, 30, 0.02)$ | $FLMA$ algorithm with $\mathcal{S}, \mathcal{L}, \mathcal{B}$ values as $10, 30, 0.02$ |
| 13 | $RP(30)$ | Reservation price algorithm ($RP$) (*El-Yaniv et al., 2001*) applied over 30 days |
| 14 | $RP^*(30)$ | Update reservation price algorithm ($RP^*$) (*Iqbal, Ahmad & Shah, 2019*) applied over 30 days |
| 15 | $KT(30)$ | Reservation price algorithm ($KT$) (*Kao & Tate, 1999*) applied over 30 days |
| 16 | $BH(30)$ | Buy and Hold algorithm applied over 30 days |

*(2010)* and *Iqbal, Ahmad & Schmidt (2012)*. Let $D_0^j$ be the initial amount of dollars at the start of a trading period $j$, and $D_T^j$ be the final amount of dollars at the end of the trading period $j$. Let, $r_j$ be the return of the $j$th trading period, then $r_j = D_T^j / D_0^j$. Assuming that there are $P$ trading periods (or trades), we define the geometric average trading period return $GPR(P)$ as;

$$GPR(P) = \left( \prod_{i=1}^{P} r_i \right)^{1/P} \tag{6}$$

Initially we do not consider any transaction fee and report our findings based on zero transaction fee. In 'How the transaction fee effect the profitablity of algorithms?', we assess the impact of the transaction fee on the returns by introducing various values of transaction fees. The transaction fees are based on coinbase—one of the popular online services dealing in buying, selling and storage of bitcoins. Coinbase charges a minimum of 1.49% transaction fee on all transactions. However, the exact value varies based on the mode of payment. For instance, for payment via credit card the transaction fee is 3.99%. We consider a set of transaction fees $TF = \{0, 2.0, 4.0\}$. We compare the geometric average period returns of the 16 strategies (see Table 2). It is also important to mention that returns are only calculated for trading periods when at least one buy transaction is followed by a sell transaction. For situations, where only buy or only sell signals are generated, no returns are taken into account.

**Table 3  Geometric average trading period returns (*GPR*) of trading strategies.**

| Strategy | Bitcoin | Euro | Yen |
|---|---|---|---|
| *VLMA*(5, 15, 0.01) | 1.021 | 1.007 | 1.003 |
| *VLMA*(5, 15, 0.02) | 1.015 | 1.043 | 1.095 |
| *FLMA*(5, 15, 0.01) | 0.986 | 1.007 | 1.005 |
| *FLMA*(5, 15, 0.02) | 1.007 | 1.04 | 1.095 |
| *RP*(15) | 1.062 | 1.009 | 1.01 |
| *RP**(15) | 1.075 | 1.01 | 1.011 |
| *KT*(15) | 0.962 | 1.001 | 1.001 |
| *BH*(15) | 0.953 | 1.0 | 1.002 |
| *VLMA*(10, 30, 0.01) | 1.033 | 1.005 | 1.018 |
| *VLMA*(10, 30, 0.02) | 1.041 | 1.021 | 1.095 |
| *FLMA*(10, 30, 0.01) | 1.032 | 1.005 | 1.018 |
| *FLMA*(10, 30, 0.02) | 1.039 | 1.02 | 1.095 |
| *RP*(30) | 1.089 | 1.013 | 1.014 |
| *RP**(30) | 1.101 | 1.014 | 1.014 |
| *KT*(30) | 0.929 | 1.0 | 0.998 |
| *BH*(30) | 0.909 | 1.001 | 1.004 |
| **Average GPR** | 1.016 | 1.012 | 1.03 |

# RESULTS AND DISCUSSIONS

In the following, we present our results from various perspectives such as the geometric average trading period returns, the number of buy/sell signals generated and the impact of the transaction fee.

## Which asset is the most profitable in terms of geometric average period return?

We calculate geometric average trading period return for each trading rule based on Eq. (6) and report our findings as shown in Table 3. It must be noted that we do not consider any transaction fee in this case. The effect of the transaction fee is discussed later in 'How the transaction fee effect the profitablity of algorithms?' It can be seen from the resultant table that the average *GPR* of the selected assets are 1.016, 1.012, and 1.03 for *BTC*, *Euro* and *Yen* respectively. Although the difference between *GPR* is not significant, *Yen* achieved a higher return than *BTC*, and *Euro*. A further analysis of the data reflects that the returns are strategy dependent as well. For instance, *RP**(30) achieved an *GPR* of 1.101 over *BTC* which is the highest returns among all the assets/strategies. Another interesting observation is the number of resultant positive and negative returns. Note that an *GPR* of at least 1 is termed as a positive return. For *BTC*, out of 16 strategies, 11 are positive. For *Euro* and *Yen*, the corresponding number of positive returns strategies are 16 and 15 respectively.

Comparing the performance of reservation price algorithms (*RP*, *RP**, *KT*), and moving average based strategies (*VLMA*, *FLMA*) with *BH*, we found that for *BTC*, the returns of all algorithms are superior than corresponding *BH* strategies. The same trend is observed

**Table 4  Highest *GPR* achieved for the assets.**

| Asset | Trading period | GPR | Algorithm |
|---|---|---|---|
| BTC | 15 | 1.075 | $RP^*$ |
| BTC | 30 | 1.101 | $RP^*$ |
| Euro | 15 | 1.432 | $VLMA(5, 15, 0.02)$ |
| Euro | 30 | 1.021 | $VLMA(10, 30, 0.02)$ |
| Yen | 15 | 1.0953 | $FLMA(5, 15, 0.02)$ |
| Yen | 30 | 1.00952 | $FLMA(10, 30, 0.02)$ |

for *Euro* except for $KT(30)$ which is inferior to $BH(30)$. All algorithms outperform $BH$ on *Yen* as well, except $KT$.

## Which strategy is the most profitable for each of the assets?

To answer the question, we analyze Table 3, and identify the best performing algorithm for each asset. We also consider the trading period length, and summarize the results in Table 4. We observed that $RP^*$ is the best algorithm for *BTC* achieving a *GPR* of 1.075 and 1.101 for trading period of length 15 and 30 respectively. For *Euro*, the corresponding algorithms are $VLMA(5, 15, 0.02)$ and $VLMA(10, 30, 0.02)$ resulting in an average *GPR* of 1.0432 and 1.021 respectively. $FLMA(5, 15, 0.02)$ and $FLMA(10, 30, 0.02)$ are the best performing algorithms for *Yen* with average *GPR* of 1.0953 and 1.0952 respectively. It is interesting to note that for each asset, a unique algorithm is adjudicated as the best performing algorithm. Further, analysis reveals that $KT$ and $BH$ are the worst performing algorithms for all the three data sets. For *BTC*, the two algorithms' returns are negative ($< 1$). For *Euro*, the returns of $KT$ and $BH$ are positive (though worst among all), and for *Yen* the returns are positive except for $KT(30)$ which is marginally less than 1. On average, $VLMA(10, 30, 0.2)$ is the best performing algorithm over all asset by achieving an average *GPR* of 1.052 which is closely followed by $FLMA(10, 30, 0.02)$. It is interesting to point out that although $RP$ and $RP^*$ assumes apriori information about the lower and upper bounds of future exchange rates, their average performance is inferior to that of $VLMA$ and $FLMA$.

In order to ensure that the performance of algorithms on *BTC* is not an anomaly, a statistical $t$-test (paired sample $t$-test) was performed with confidence level of 95% ($p \leq 0.05$). The tests were performed on the returns of algorithms for *BTC* considering 15 and 30 days trading duration. Tables 5 and 6 summarizes the results of paired $t$-test for the returns of *BTC* on various algorithms for 15 and 30 days trading periods. Recall from Table 3 that $RP^*$ is the best performing algorithm for *BTC*. Table 5 confirms that with 95% confidence the improved performance of $RP^*$ over all algorithms (except $VLMA(5, 15, 0.02)$, and $FLMA(5, 15, 0.02)$) is not by chance. For $VLMA(5, 15, 0.02)$, and $FLMA(5, 15, 0.02)$ the confidence level is still significant ( 93% and 94%). Other than $RP^*$, and $RP$ no other algorithm exhibits a significant confidence in the returns over *BTC*. However, except for $FLMA(5, 15, 0.01)$ and $KT(15)$ all other algorithms have shown the potential to beat the market, i.e., the returns are better (and statistically significant) than $BH$ strategy. For 30 days trading period, the returns of moving average based strategies

**Table 5 Paired sample $t$-test for the returns of *BTC* with confidence interval of 95% (15 days trading period).**

| Algorithms | VLMA(5,15,0.02) | FLMA(5,15,0.01) | FLMA(5,15,0.02) | RP(15) | RP*(15) | KT(15) | BH(15) |
|---|---|---|---|---|---|---|---|
| VLMA(5,15,0.01) | 0.857 | 0.419 | 0.982 | 0.07 | 0.026 | 0.083 | 0.023 |
| VLMA(5,15,0.02) | – | 0.317 | 0.848 | 0.145 | 0.072 | 0.053 | 0.012 |
| FLMA(5,15,0.01) | – | – | 0.346 | 0.037 | 0.015 | 0.338 | 0.175 |
| FLMA(5,15,0.02) | – | – | – | 0.127 | 0.061 | 0.089 | 0.041 |
| RP(15) | – | – | – | – | 0 | 0 | 0 |
| RP*(15) | – | – | – | – | – | 0 | 0 |
| KT(15) | – | – | – | – | – | – | 0.944 |

**Table 6 Paired sample $t$-test for the returns of *BTC* with confidence interval of 95% (30 days trading period).**

| Algorithms | VLMA(10,30,0.02) | FLMA(10,30,0.01) | FLMA(10,30,0.02) | RP(30) | RP*(30) | KT(30) | BH(30) |
|---|---|---|---|---|---|---|---|
| VLMA(10,30,0.01) | 0.541 | 0.425 | 0.574 | 0.559 | 0.393 | 0.054 | 0.043 |
| VLMA(10,30,0.02) | – | 0.471 | 0.137 | 0.758 | 0.572 | 0.063 | 0.042 |
| FLMA(10,30,0.01) | – | – | 0.499 | 0.486 | 0.329 | 0.053 | 0.043 |
| FLMA(10,30,0.02) | – | – | – | 0.73 | 0.546 | 0.065 | 0.43 |
| RP(30) | – | – | – | – | 0.008 | 0 | 0 |
| RP*(30) | – | – | – | – | – | 0 | 0 |
| KT(30) | – | – | – | – | – | – | 0.616 |

are statistically significant than *BH* only (see Table 6), whereas *RP\** achieves statistically significant returns than *RP*, *KT*, and *BH* only.

## How the number of buy and sell signals vary for *BTC*?

We record the number of buy and sell signals, as well as the number of completed transactions. A transaction is completed when for a buy signal the corresponding sell transaction occurs. Figure 1 summarizes the number of buy, sell and completed transactions.

We observed that considering 15 days trading period for *BTC*, *VLMA* and *FLMA* based strategies resulted in 6% completed transactions. *VLMA* generates 2% more buy and sell signals than *FLMA*. This is logical as *VLMA* based strategies do not have any holding period and are free to generate a signal if the corresponding criterion is met. For reservation price algorithms, the number of completed transactions are in the range of $27 - 32\%$. Buy and hold has the highest number of completed transactions as it does not generate buy and sell signals based on some predefined criterion, but instead buys on the first trading day and sells on the last trading day irrespective of the offered exchange rate. For trading period of length 30 days, the same trend is observed for *VLMA* and *FLMA* based strategies.

Our analysis of the data reveals that for *VLMA* on average 96% of the buy transactions remains open whereas the corresponding number for the sell signal is 94%. Likewise, the percentage of open buy and sell signals for *FLMA* are 22% and 19% only. For reservation price algorithms, the number of completed transactions are reduced to $14 - 20\%$.

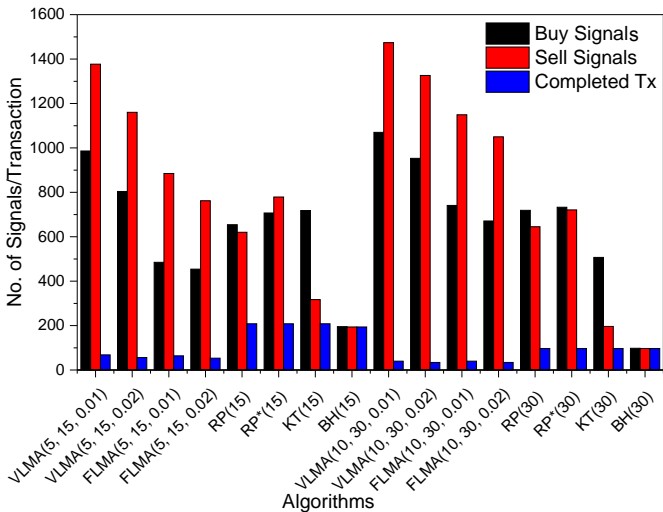

**Figure 1**  **Number of Buy and Sell Signals and Complete Transactions for BTC.**

### What are the number of positive and negative returned transactions for *BTC*?

We investigate completed transactions from the perspective of positive vs negative returns for *BTC*. We define a transaction to yield positive returns if the sell price is higher than the buy price, i.e., $r_j > 1$ where $r_j$ represents the returns of the trading period $j$.

For 15 days trading period, we observe that the *VLMA* and *FLMA* based strategies have higher percentage ($> 50\%$) of negative returned transactions. For the reservation price policy, *KT* has more negative transactions ($> 50\%$) whereas *RP* and *RP\** have higher positive transactions. *RP* has 66.8%, and *RP\** has 72 positive returned transactions. *BH* has 45% positive returned transactions. The worst rate of positive returned transaction is observed for *KT* (40%).

For trading duration of 30 days, rather surprisingly, the percentage of positive returned transaction increased slightly for *VLMA*, *FLMA* and *KT*, whereas a reduction is observed for *RP* and *RP\**. Figure 2 is a graphical summary of the positive and negative returned transaction for *BTC*.

### How the transaction fee effect the profitablity of algorithms?

Transaction fee can be a vital factor in the profitability of any trading algorithm. We consider a transaction fee $TF = \{0\%, 2\%, 4\%\}$ and calculate *GPR* to find the effect on the profitability. Figure 3 is a graphical representation of the effect of transaction fee on *GPR* of algorithms for *BTC*. We observed an average reduction of 3.9% and 7.6% in the *GPR* of algorithms when transaction fee of 2% and 4% are levied. Introducing a transaction fee of 2% reduced the positive returned strategies from 11 to 5 only, which are further reduced to 2 (*RP* and *RP\**) when the transaction fee is increased to 4%. For *Euro*, the profitability of algorithm is severely reduced from 16 to 0 strategies when the transaction fee of 4% is introduced. Rather interestingly, for *Yen* the introduction of transaction fee reduces

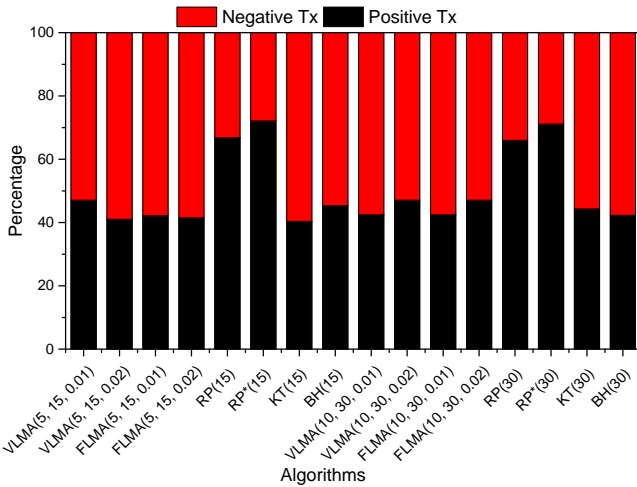

**Figure 2   Positive vs negative returned transactions.**

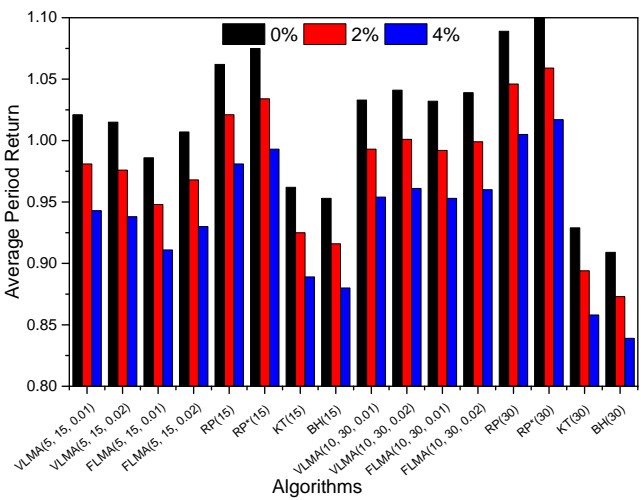

**Figure 3   The effect of transaction fee on *GPR* of *BTC*.**

the number of profitable strategies from 15 to 4 after introduction of 2% transaction fee. However, there is no change in the number of profitable strategies when the transaction fee is increased to 4%. For *Yen*, the four profitable strategies are *VLMA* (5, 15, 0.02), *FLMA* (5, 15, 0.02), *VLMA* (10, 30, 0.02), *FLMA* (10, 30, 0.02). This also reflects that for variable length moving average strategies to be profitable the band value is vital. For smaller band values, the strategies might not be profitable.

## CONCLUSION

We evaluated the short term profitability of *BTC* over a set of reservation price and moving average based algorithms against the euro and the yen for a period of 8 years. Based on the

average *GPR*, *BTC* seems less profitable venture than the yen; however, a deeper analysis revealed that the answer the profitability is strategy dependent as well. *RP\** achieved an average *GPR* of 10% for a trading period of 30 days, which is the maximum return obtained by any trading algorithm among the three assets. This confirmed *BTC* as an attractive investment opportunity for a short term investment. Our analysis also revealed that *RP* and *RP\** are the best performing algorithms on *BTC*, whereas moving average based algorithms return higher profits for the euro and the yen. It is also shown that the selected set of algorithms beat the buy and hold approach except on the yen where the returns of *KT* are less than that of buy and hold. Further, we highlighted that the returns of all the selected algorithms became negative except for *RP* and *RP\** when a transaction fee of 2% was introduced. Increasing the transaction fee to 4% resulted in positive returns for *RP* and *RP\** on 30 days investment horizon. For all other algorithms and their variants the returns were negative for a transaction fee of 4%.

To the best of our knowledge, this study is the first of its kind to evaluate the profitability of *BTC* using a set of trading algorithms and against fiat currencies. Future work can include finding an optimized portfolio of fiat and crypto-currencies for short and long term investment.

### Funding
The authors received no funding for this work.

### Competing Interests
The authors declare there are no competing interests.

### Author Contributions
- Iftikhar Ahmad conceived and designed the experiments, performed the experiments, performed the computation work, authored or reviewed drafts of the paper, and approved the final draft.
- Muhammad Ovais Ahmad conceived and designed the experiments, analyzed the data, authored or reviewed drafts of the paper, and approved the final draft.
- Mohammed A. Alqarni conceived and designed the experiments, prepared figures and/or tables, authored or reviewed drafts of the paper, and approved the final draft.
- Abdulwahab Ali Almazroi performed the experiments, analyzed the data, performed the computation work, prepared figures and/or tables, authored or reviewed drafts of the paper, and approved the final draft.
- Muhammad Imran Khan Khalil performed the experiments, analyzed the data, performed the computation work, prepared figures and/or tables, authored or reviewed drafts of the paper, and approved the final draft.

### Data Availability
Data and code can found in the Supplementary Files.

## Supplemental Information

Supplemental information for this article can be found online at http://dx.doi.org/10.7717/peerj-cs.337#supplemental-information.

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
