# Peer review of "Using algorithmic trading to analyze short term profitability of Bitcoin"

_PeerJ Computer Science, doi:10.7717/peerj-cs.337_

## Round 0.1 · original submission · Minor Revisions

After considering the reviews made by the reviewers, there are some aspects regarding the methodological aspects and presentation of results that need to be addressed. Please, resubmit the paper after incorporating the above-mentioned improvements.

Reviewer 1 ·

Basic reporting

no comment

Experimental design

A weakness of the manuscript is the unclear objective of the paper. 5 research questions are presented without explanation why these questions are relevant. So it is not clear if the paper is about to find the best algorithm for a specific asset (BTC) or to compare profitability of specific assets using the algorithms.

Please do more explanation why you use this 5 research quesitions. And make the objective of the paper clearer.

Validity of the findings

no comment

Additional comments

1. The algorithms 1 – 4 generate buy and sell signals even if it not possible to buy or sell the assets. Please change your algorithms in a way that they consider previous prices and only generate buy or sell signals, if it is possible to buy respectively sell. You cannot sell without buying before. Specially for algorithm 4 you even describe, that buy and sell signals are generated if two moving averages cut each other (lines 192 – 195). Nevertheless, in algorithm 4, signals are generated, if one moving average is higher/lower than the other.
2. In line 206 – 208 you describe that algorithm 4 generate signals without a cut of the two moving averages. Please be consistent.
3. In this context, table 2 should be adjusted. Here you show the number of signals which are generated over the whole data set per algorithm. There is a large number of buy and sell signals but the number of completed transactions is very small. It seems you assume that if the last day of a trading period occurs, it is not necessary to sell all units bought before. This is a typical assumption in the literature of competitive algorithms which you refer to. Note, if there is no need to sell the units which are bought before, your benchmark (the buy and hold strategy) is in disadvantage. Specially, if transactions costs are considered.
4. Further it is not clear, if you take the return of trading periods into account in which only buy signals are generated.
5. If all bought units have to be sold at the last day of a trading period, this should also be noted in algorithms 1 – 4.
6. Why is FLMA the only algorithm which is not presented?
7. Another important issue with your manuscript is the relevance of your research questions. The five research questions are listed in the lines 141 – 145 without any explanation why these questions are relevant.
8. One unanswered question is how do you determine m and M for RP?
9. Figure 2 shows the transactions per algorithm. Figure 1 shows the effect of transactions costs. Because the results of figure 1 depend on the findings of figure 2, I would change the order of figures and show figure 2 first. Further, this change in order gives you the opportunity to discuss the findings of figure 1. This could be very helpful, because at the first sight it seems, that the algorithms with a low number of transactions, haven’t a lower impact from transaction costs.
10. Table 2 is a summary of the selected scenarios but not of the algorithms. In fact, you use six algorithms, in two different settings (15 days, 30 days). Further, it is not clear, why you choose different parameters for the algorithms VLMA and FLMA for 15 days and 30 days. Why do you not use the four combinations of parameter for 15 and 30 days?
11. In line 203 you define e_t again, without using it after that.
12. Please define the first two variables in line 209 before using it.
13. Please define the variables i and f before using it. Line 226 -228. Why are you not using T?
14. Please define the variable T before using it. (First use in algorithm 2)
15. Please define the variable q_t before using it. (First use in algorithm 2). Do you mixed up q_t and e_t?

Reviewer 2 ·

Basic reporting

no comment

Experimental design

no comment

Validity of the findings

no comment

Additional comments

The authors investigated the short term profitability of Bitcoin cryptocurrency against two fiat currencies, namely Euro and Yen.
The analysis was conducted implementing both reservation price algorithms and moving average based trading techniques over an eight year period, ranging from January 1, 2011 to 31 December, 2018.
The results were compared with those obtained with the classical buy-and-hold trading technique and were evaluated in terms of the (Geometric) average trading period return criterion.
This study highlights that on average, in short term Yen is more profitable than Bitcoin and Euro, although the answer also depends on the choice of the algorithm.

The article is clearly written and the analyses seem to be performed rigorously. Raw data and code are provided by the authors and easily accessible. For these reasons, I think that the paper would be of interest for the readership of PeerJ Computer Science.
However, there are some minor changes that must be addressed for the article to be ready for publication.

The "Introduction" of the paper should be proofread. I suggest to move the content of section 2 (Bitcoin - A brief overview) to the "Introduction" section, describing in a more detailed way the Blockchain technology, focusing more on its main feature of public ledger. The authors made a comparison between Bitcoin cryptocurrency and two fiat currencies, namely Euro and Yen. I think that they didn't highlight the main problem of this comparison, which is also the main feature that distinguish their markets, i.e. fiat currencies depends only on classical macroeconomic variables while cryptocurrencies are virtual currencies based on Blockchain technology and therefore their market also depends on variables related to the technology itself. Please go into this in more detail in the "Introduction" section.
Figures 1 and 2 shows the results obtained for the Bitcoin cryptocurrency. If there are no length limits, why not also report the figures for Euros and Yen? Please, report all results.

The purpose of this paper is to ascertain the short term profitability of Bitcoin comparing it with those of the Euro and Yen fiat currencies. This comparison gives a sense of how different BTC is. Actually, to make this analysis more robust I would include the analysis of another cryptocurrency in order to have an equal comparison, for example (Bitcoin, Ethereum) VS (Euro, Yen). The choice of BTC and ETH is quite straightforward, since they are the two most valuable cryptocurrency at the moment. Please, expand the experiment with one more cryptocurrency.
- Page 4, table 1. I suppose the data are in USD, but please always report the unit of measure or specify it in the table description.
- Page 7, section 5.2. The authors should be more accurate in the description of the chosen evaluation criterion. Equation 6 doesn't represent the Average Trading Period Return, which is the simple arithmetic mean of returns, but instead the Geometric Average Trading Period Return, that is the geometric mean of returns. Furthermore, the authors should explain how this evaluation criterion should be use in this study. For example, when dealing with time series prediction we use MAE (Mean Absolute Error) error to evaluate results. A lower MAE means better performance, while a worse MAE means worse results. Please, clarify these points.
- Page 7, lines 221-224, line 235. Page 9, line 285. The authors should explain how and why they choose this values. Did the authors use or follow a particular criterion for the choice of these values? For example, when dealing with clustering the number of cluster to build is usually chosen because of the Elbow method. This is a critical issue that the authors should clarify.
- Page 11, Figure 2. Please, choose "Signal" or "Signals".
- Page 11, line 330. "whereas moving based algorithms...". Is "whereas moving average based algorithms..." what the authors would like to say?

---

## Round 0.2 · accepted · Accept

The reviewers consider their comments as properly addressed therefore your manuscript can be accepted for publication.

Reviewer 1 ·

Basic reporting

no comment

Experimental design

no comment

Validity of the findings

no comment

Additional comments

Thank you for taking so many of my improvement suggestions into account.

I have no further issues with the manuscript.

Good Work!

Reviewer 2 ·

Basic reporting

No comment

Experimental design

no comment

Validity of the findings

no comment

Additional comments

The authors have provided a clear and detailed explanation of all the points raised. For this reason I believe that the article is now ready for publication.

---

## Author Rebuttal · Round 0.2

# Reply to Reviewer's Comments

We are thankful to the reviewers for the constructive feedback and reviews directed to improve the quality of our manuscript. We have carefully considered all the suggestions and comments provided by the reviewers and have revised the paper accordingly.

In the following, we provide a point-by-point reply to reviewers' comments.

## Reviewer 01:

**Comment 1**: *The algorithms 1 – 4 generate buy and sell signals even if it not possible to buy or sell the assets. Please change your algorithms in a way that they consider previous prices and only generate buy or sell signals, if it is possible to buy respectively sell. You cannot sell without buying before. Specially for algorithm 4 you even describe, that buy and sell signals are generated if two moving averages cut each other (lines 192 – 195). Nevertheless, in algorithm 4, signals are generated, if one moving average is higher/lower than the other.*

**Reply**: Thank you for pointing out the confusion. Although the algorithms generate buy and sell signals whenever the condition is met, it does not necessarily result in buy and sell transactions. Note that, first a buy transaction must occur, and only then a sell transaction can take place. If there are any subsequent buy signals after the first sell signal, the algorithm will ignore them. The same rule applies for a sell transaction.

**Comment 2**: *In line 206 – 208 you describe that algorithm 4 generate signals without a cut of the two moving averages. Please be consistent.*

**Reply**: Thank you for highlighting the confusion. We have changed the text in order to be consistent with the earlier text.

**Comment 3**: *In this context, table 2 should be adjusted. Here you show the number of signals which are generated over the whole data set per algorithm. There is a large number of buy and sell signals but the number of completed transactions is very small. It seems you assume that if the last day of a trading period occurs, it is not necessary to sell all units bought before. This is a typical assumption in the literature of competitive algorithms which you refer to. Note, if there is no need to sell the units which are bought before, your benchmark (the buy and hold strategy) is in disadvantage. Specially, if transactions costs are considered.*

**Reply**: We assume that you are referring to Fig 2 instead of Table 2. The comment is indeed very thoughtful regarding the assumption of the last day. Luckily, we have considered the scenario, and the algorithms are forced to sell on the last day trading day. Therefore, the buy and hold strategy is not at disadvantage. The same is reflected in the revised description of algorithms.

**Comment 4**: *Further it is not clear, if you take the return of trading periods into account in which only buy signals are generated.*

**Reply**: Again, this is a thoughtful comment, and we have already considered this in our experiments. However, we have now added the following text in our manuscript to remove any confusion pertaining to the point.

> "It is also important to mention that returns are only calculated for trading periods when at least one buy transaction is followed by a sell transaction. For situations, where only buy or only sell signals are generated, no returns are taken into account."

**Comment 5**: *If all bought units have to be sold at the last day of a trading period, this should also be noted in algorithms 1 – 4.*

**Reply**: Changes are made in Algorithms 1-4.

**Comment 6**: *Why is FLMA the only algorithm which is not presented?*

Reply: As FLMA has the same working principle as VLMA except for the holding period, therefore, to avoid redundancy, the algorithm is not repeated. However, we have modified the text to clearly mention the point. Following is the modified text;

> "$FLMA$ works on the same principle as stated in Algorithm~\ref{AlgoVMA}. However, $FLMA$ differs from the $VLMA$ by introducing a holding period, i.e., once a signal is generated then the position must be held for a fixed number of days. Any signal generated during the holding period is ignored."

**Comment 7**: *Another important issue with your manuscript is the relevance of your research questions. The five research questions are listed in the lines 141 – 145 without any explanation why these questions are relevant.*

**Reply**: Thank you for highlighting the point. We have added the following text to relate our research questions with historical literature.

> "Note that the research questions are not arbitrarily but are instead rooted in the literature. For instance, $RQ1$ is based on \cite{hsu2016} who used Japanese yen, German mark/euro, U.K. pound, and Swiss franc as base currency in their study and evaluated the profitability of technical trading rules. Likewise, $RQ2$ is variant of research question posed in \cite{abbey2012technical}. In \cite{abbey2012technical} the authors examined if technical trading rules can be profitable for individual traders. In the similar manner, $RQ5$ is studied by a number of researchers including~\cite{hsu2016} and \cite{Ahmad2012}."

**Comment 8**: *One unanswered question is how do you determine m and M for RP?*

**Reply**: It is assumed that these estimated values are known to the algorithm in advance. This is the key assumption of the algorithm, as otherwise, it will not be possible to derive a competitive algorithm. The same is stated in the manuscript as well.

> "~\cite{YFKT01} assumed a priori information about the lower (minimum possible price $m$) and upper (maximum possible price $M$) bound of prices, and presented a reservation price algorithm."

**Comment 9**: Figure 2 shows the transactions per algorithm. Figure 1 shows the effect of transactions costs. Because the results of figure 1 depend on the findings of figure 2, I would change the order of figures and show figure 2 first. Further, this change in order gives you the opportunity to discuss the findings of figure 1. This could be very helpful, because at the first sight it seems, that the algorithms with a low number of transactions, haven't a lower impact from transaction costs.

**Reply**: Thank you for the insight. In order to accommodate the comment, we have changed the order of research questions and subsequently, we have re-ordered the sub-section in Section 6 as well.

**Comment 10**: *Table 2 is a summary of the selected scenarios but not of the algorithms. In fact, you use six algorithms, in two different settings (15 days, 30 days). Further, it is not clear, why you choose different parameters for the algorithms VLMA and FLMA for 15 days and 30 days. Why do you not use the four combinations of parameter for 15 and 30 days?*

**Reply**: Thank you for highlighting the inconsistency. We have changed the caption of Table 2 accordingly.
Currently we are using 4 variants of VLMA and FLAM respectively, the remaining 8 combinations are not included as we are focusing on the short-term investment, including the combination such as short-term average over 5 days and long-term average over 30 days might not be suitable to justify.

**Comment 11**: *In line 203 you define e_t again, without using it after that.*

**Reply**: The redundant definition of e_t is deleted.

**Comment 12**: *Please define the first two variables in line 209 before using it.*

**Reply**:  Thank you for pointing out the discrepancy. The variables are now defined.

**Comment 13**: *Please define the variables i and f before using it. Line 226 -228. Why are you not using T?*

**Reply**: The text is modified to make it self-explanatory. The revised text is as following;

"Let $D_{0}^{j}$ be the initial amount of dollars at the start of a trading period $j$, and $D_{T}^{j}$ be the final amount of dollars at the end of the trading period $j$. Let, $r_j$ be the return of the $j$th trading period, then $r_j = D_{T}^{j}/D_{0}^{j}$."

**Comment 14**: *Please define the variable T before using it. (First use in algorithm 2).*

**Reply**: Thank you for pointing out the missing variable definition. The following text is added.

"$T$ represents the number of days in a trading period, … "

**Comment 15**: *Please define the variable q_t before using it. (First use in algorithm 2). Do you mixed up q_t and e_t?*

**Reply**: Thank you for pointing out the inconsistency. Indeed e_t should have been used instead of q_t. The error is corrected.

## Reviewer 02:

**Comment 1**: *The "Introduction" of the paper should be proofread. I suggest to move the content of section 2 (Bitcoin - A brief overview) to the "Introduction" section, describing in a more detailed way the Blockchain technology, focusing more on its main feature of public ledger.*

**Reply**: Thank you for the suggestion. Section 2 is now merged in Section 1.

**Comment 2**: *The authors made a comparison between Bitcoin cryptocurrency and two fiat currencies, namely Euro and Yen. I think that they didn't highlight the main problem of this comparison, which is also the main feature that distinguish their markets, i.e. fiat currencies depends only on classical macroeconomic variables while cryptocurrencies are virtual currencies based on Blockchain technology and therefore their market also depends on variables related to the technology itself. Please go into this in more detail in the "Introduction" section.*

**Reply**: Thank you for a great suggestion. We have added the following text about differentiating properties of cryptocurrencies and fiat currencies in the Introduction.

"Beside similarities, such as the price regulation based on demand and supply, there are some key differences between fiat and crypto currencies like $BTC$. For instance, $BTC$ has no centralized authority (like the Federal Reserve) that controls the supply,

i.e., $BTC$ and by extension all cryptocurrencies are decentralized by nature. The value of a fiat currency is generally dependent on factors such as inflation rate in a country, the interest rates, balance between import and export and monetary policy. In contrast, the value of $BTC$ can be determined by several factors such as transactional demand, media speculation, buzz around the technology, and acceptability etc~\citep{nguyen2018factors,wang2017buzz}. Other differentiating aspects include legality, tangibility, and storage."

**Comment 3**: *Figures 1 and 2 shows the results obtained for the Bitcoin cryptocurrency. If there are no length limits, why not also report the figures for Euros and Yen? Please, report all results.*

**Reply**: The comment is valid, however, as the results are not interesting and somehow similar to the BTC, therefore, we omitted it. This also helped in keeping focus on BTC results.

**Comment 4**: *The purpose of this paper is to ascertain the short-term profitability of Bitcoin comparing it with those of the Euro and Yen fiat currencies. This comparison gives a sense of how different BTC is. Actually, to make this analysis more robust I would include the analysis of another cryptocurrency in order to have an equal comparison, for example (Bitcoin, Ethereum) VS (Euro, Yen). The choice of BTC and ETH is quite straightforward, since they are the two most valuable cryptocurrency at the moment. Please, expand the experiment with one more cryptocurrency.*

**Reply**: Thank you for the valuable suggestion. As the learned reviewer has pointed out the purpose of this work is to ascertain the short-term profitability of BTC against fiat currencies, therefore we restrained from including other cryptocurrencies such as ETH. In fact, we believe that adding ETH to the mix will compromise the clarity and focus of the work. Therefore, we put forth our request to not include ETH in the comparison.

**Comment 5**: *Page 4, table 1. I suppose the data are in USD, but please always report the unit of measure or specify it in the table description.*

**Reply**: The statistics have various units and some entries do not have a unit at all. For instance, for number of observations, we cannot use BTC, USD, Euro or Yen as unit. The same applies for standard deviation, skewness, kurtosis and order of correlation as well. However, it is mentioned in the text that "a single data point represents the amount of currency that can be purchased by spending 1 US$", therefore, the same can be used for Table 1 when reading rows about Minimum, Maximum and Mean values.

**Comment 6**: *Page 7, section 5.2. The authors should be more accurate in the description of the chosen evaluation criterion. Equation 6 doesn't represent the Average Trading Period Return, which is the simple arithmetic mean of returns, but instead the Geometric Average Trading Period Return, that is the geometric mean of returns. Furthermore, the authors should explain how this evaluation criterion should be use in this study. For example, when dealing with time series prediction we use MAE (Mean Absolute Error) error to evaluate results. A*

*lower MAE means better performance, while a worse MAE means worse results. Please, clarify these points.*

**Reply**: Thank you for highlighting an important point. We have modified the definition accordingly and changed all instance in the text as well.

Further, it is pertinent to mention that Geometric Average Period Return is one of the standard mechanisms to evaluate the performance of trading algorithms. For reference, we refer to Schmidt et al. (Electronic Notes in Discrete Mathematics 36 (2010) 519–526). Mohr et al. (doi.org/10.1016/j.sorms.2014.08.001), and Iqbal et al. (10.4230/OASIcs.SCOR.2012.43). We have also added the reference in the text at appropriate location (in subsection Evaluation Criterion)

**Comment 7**: Page 7, lines 221-224, line 235. Page 9, line 285. The authors should explain how and why they choose this values. Did the authors use or follow a particular criterion for the choice of these values? For example, when dealing with clustering the number of cluster to build is usually chosen because of the Elbow method. This is a critical issue that the authors should clarify.

**Reply**: Thank you for pointing out an important aspect. As we are dealing with short term profitability, therefore, we used 5 and 10, for short term moving average, 15 and 30 for long term moving averages. The band values are selected based on the literature review (doi.org/10.1016/j.rfe.2013.05.004) and (http://www.jstor.org/stable/2328994). We have added the references in the text as well.

**Comment 8**: *Page 11, Figure 2. Please, choose "Signal" or "Signals".*

**Reply**: Thank you for identifying the inconsistency, we have corrected it.

**Comment 9**: *Page 11, line 330. "whereas moving based algorithms...". Is "whereas moving average based algorithms..." what the authors would like to say?*

**Comment**: Thank you for highlighting the missing word. The changes are made.